# Research on Renewable Energy Trading Strategies Based on Evolutionary Game Theory

Fei Huang [1], Hua Fan [1], Yunlong Shang [1], Yuankang Wei [1], Sulaiman Z. Almutairi [2,*], Abdullah M. Alharbi [3], Hengrui Ma [4,5] and Hongxia Wang [4,5,6]

[1]   Guangxi Power Exchange Center Co., Ltd., Nanning 530023, China; huangf1502@126.com (F.H.);
    fan_h.dd@gx.csg.cn (H.F.); sun15735513996@163.com (Y.S.); weiyuankang@163.com (Y.W.)
[2]   Department of Electrical Engineering, College of Engineering, Prince Sattam bin Abdulaziz University,
    Al Kharj 16278, Saudi Arabia
[3]   Department of Electrical Engineering, College of Engineering in Wadi Addawasir, Prince Sattam bin
    Abdulaziz University, Wadi Addawasir 11991, Saudi Arabia; am.alharbi@psau.edu.sa
[4]   Hubei Engineering and Technology Research Center for AC/DC Intelligent Distribution Network,
    School of Electrical Engineering and Automation, Wuhan University, Wuhan 430072, China;
    henry3764@foxmail.com (H.M.); 2018282070092@whu.edu.cn (H.W.)
[5]   School of Electrical and Automation, Wuhan University, Wuhan 430072, China
[6]   Department of Electrical & Computer Engineering, University of Denver, Denver, CO 80208, USA
*   Correspondence: s.almutairi@psau.edu.sa

**Abstract:** The authors construct a tripartite evolutionary game model that considers renewable energy, traditional coal-fired power plants, and market users. We propose multiple income matrices under different strategies, conduct evolutionary stability analysis, and form a series of assumptions that meet the stability of the game. We also simulate and analyze the impact of key factors—such as assessment costs, different pricing behaviors of coal-fired power plants, electricity prices of renewable energy, and green electricity demand—on the stability of the game. In addition, the market equilibrium points that can be achieved by optimizing trading strategies and their optimization status in promoting renewable energy consumption are analyzed. Based on the operational characteristics of the Guangxi electricity market in China and the trading situation of renewable energy, an evolutionary game method is applied to conduct empirical research. The trading behavior and evolution of all parties in the market are fully analyzed and are then applied to the construction and mechanism improvement of the electricity market.

**Keywords:** evolutionary game; electricity market; stability analysis; renewable energy consumption

## 1. Introduction

### 1.1. Background of Renewable Energy Development

Renewable energy installation and electricity generation are growing rapidly globally, with solar and wind power expanding as technological advances and cost reductions make these energy sources more viable and economical. Renewable energy sources are expected to continue to grow in the future, resulting in a substantial increase in their share of the global energy mix as policy support, technological innovation, and concerns about climate change deepen. In the long run, the realization of more efficient energy conversion, storage, and distribution systems and the rapid development of renewable energy will drive future energy development trends.

With the "dual-carbon" goal of China, the power industry must reduce carbon emissions; hence, low-carbon transformation is particularly important. As a result, the role of the electricity-market-oriented trading platform in achieving carbon and emissions reduction has become critical.

Renewable-energy-market-oriented consumption policy has been introduced on multiple levels. Nationally, the government has directed the establishment of a sound renewable

energy consumption guarantee mechanism and other related policies, which establish energy consumption limits and goals for each province, consumption modes, and related safety measures. At the regional and local levels, Guangdong, Guangxi, Yunnan, Guizhou, and Hainan jointly released directives on renewable energy power consumption for these five southern provinces, including basic rules for trading, settlement, and information disclosures.

Since 2022, the installed renewable energy capacity in Guangxi has seen a substantial increase through increasingly diversified trading channels. Guangxi's power market annual trading program cleared more than 1800 h of wind power for participation in the market, expanding the space of renewable energy through market-based consumption. Wind power and other renewable energy sources have also gradually entered the market, which will allow Guangxi's power-market-oriented transactions to have a greater impact on the traditional coal-fired power generation enterprises and renewable energy consumption assessment of the weight of power users and other market players, and transaction decision-making behavior will change.

### 1.2. Evolutionary Game Theory Model

Game theory is a mathematical method that can be utilized to solve complex problems. Evolutionary game theory is one of its branches, in which the main conditions are based on the limited rationality of the parties to the game and the infinity of the game subject, through the evolutionary model between individuals similar to the natural selection of superiority and inferiority of the survival of the fittest, to achieve the state of equilibrium in individual learning, imitation, and an iterative game process, so as to depict the group's gaming behavior. Therefore, this method has been widely used in engineering, economics, and other fields. The main approaches to the study of market trading behaviors at the current stage include experimental economics, intelligent reinforcement algorithms based on deep learning models, and evolutionary game theory.

As the name suggests, evolutionary stability in game theory, referred to as "evolutionarily stable strategy" (ESS), is a method for describing the behavior of a group. ESS is closely related to biological modeling and is informed by drawing on the influence of variation in large populations in genetics to form the game system. The main idea is to regard the parties to the game as having a limited rationality and to constantly adjust the game process dynamically, instead of through the use of traditional game assumptions of completely rational people. This is considered to be a more suitable method for modeling the reality of complex economic and engineering conditions.

Evolutionary game theory has become a hot research area in the field of electricity markets with its limited information and limited rationality assumption. For example, Gao Jie and Sheng Zhaohan analyzed the bidding process of power generation enterprises and simulated a reasonable and stable bidding strategy for power generation utilizing ESS [1]. Wu Ning, Zubo, analyzed and researched the bidding rules of power plants in the market-based trading process and used evolutionary game theory to measure stability in the offer bidding stage [2]. Xinru Li constructed a power generation model to simulate the bidding strategy of power plants using evolutionary game theory to form the change of offer strategy under the influence of different policies [3]. Gao Jie similarly used evolutionary game theory as the basis of bidding rules for the electricity market, to conduct an analytical study of market bidding behavior; the study emphasized the impact of policy regulation on market behavior [4]. Extending ESS to the study of multi-body behavioral strategies in the electric power market, Cheng Lefeng and Yu Tao utilized asymmetric evolutionary game modeling to simulate the stability decision-making of various types of market bodies and to completely analyze the dynamic characteristics under a variety of typical transaction scenarios. Yang Zhao simulated the various types of behavior in the power market as a three-party subject transaction game, constructing an evolutionary replication equation with offering and selling as the main body, to achieve the optimal solution to maximize interests in the transaction process. This simulation was carried out using the East China power market#.

### 1.3. Purpose and Significance of the Study

As power market reform continues to evolve, market mechanisms continue to be improved. In addition, wind power, photovoltaic, and other renewable energy sources will enter the power market on a larger scale through the market-based approach to consumption. This will inevitably impact the power market and the trading strategies employed by existing market players. In this paper, by simulating the different trading behaviors of coal-fired power plants, renewable energy power plants, and market users, we construct a revenue matrix of each of these components, establish replicated dynamic equations, form the Jacobi matrix, solve the eigenvalues, and analyze the assumptions on the formation conditions of each stable point, presenting research on clean energy through market-based consumption situations, coal-fired power plants, renewable energy, users, and other three-party game strategies.

### 1.4. Literature Review

The authors utilized references [5–7] to review the current state of theoretical research and research progress related to emissions trading. In recent years, research on emissions trading has gradually introduced game theory, information economics, system planning, and auction theory, with the main focus being on environmental externalities and transaction costs. Domestic research has primarily focused on the applicability of policies, initial allocation, auction pricing methods, and other aspects. At present, scholars have used methods such as the Bass diffusion model [8], the Learning Curve Theory [9], system dynamics [10,11], and game theory [12] to solve this problem. Reference [8] used the Bass model to predict the trend of wind power diffusion in coupled energy systems. Reference [9] used a learning curve to study the impact of onshore wind and photovoltaic power generation investments on future installed capacity. The diffusion problem of distributed photovoltaic technology guided by electricity prices based on system dynamics modeling ideas was studied by [10]. In [12], the authors analyzed the impact of fixed electricity prices and grid subsidy policies on the development of renewable energy using the Stackelberg game method. In each of these studies, the authors generally focused on studying the impact of factors such as price, subsidy policy, and investment return on the development of renewable energy from a macro perspective, which rarely involves analyzing the complexity of complex game processes and development decisions within user groups from a micro perspective on the promotion and trading of renewable energy.

Renewable energy trading policies involve complex game processes within user groups. Evolutionary game theory, as an important branch of game theory, advocates bounded rationality among game entities and has clear advantages when explaining evolutionary mechanisms [13–15] to provide new proposals for the study of the long-term evolution of renewable energy development from the perspective of multi-agent micro decision-making [16–19]. In [17], the authors constructed an evolutionary game model incorporating the government, rooftop owners, power grids, and photovoltaic enterprises to study the impact of different subsidy policies on the promotion of household rooftop photovoltaic systems in the entire county. In [18], a multi-party evolutionary game model was constructed for the government, power generation enterprises, and power grid companies to analyze the factors affecting the implementation of renewable energy quota policies. With a focus on the game relationship between thermal power manufacturers, new energy manufacturers, and power users, [19] used evolutionary game theory to analyze the impact of policy standards, market prices, and other factors on the long-term strategy choices of all parties in the game. However, the existing research assumes an ideal scenario for conducting random repeated games between multiple agents. In reality, game agents have complex topological and statistical characteristics in social systems, and their interactions are not fully coupled or completely random.

Complex network theory, as a mathematical method that focuses on the analysis of connection relationships, can effectively describe the complex relationships between agents. Currently, a large number of studies have applied complex network theory to the evolution

analysis of complex systems [20–23]. For example, [21] constructed a complex network based on the relationship between electricity consumption characteristics of power users to construct an evolutionary analysis of electricity consumption characteristics of power users under different factors. In [22], complex network theory was used to analyze the evolution and morphological characteristics of regional energy networks based on urban growth characteristics. Similarly, [23] considered the complex connections between electric vehicle charging stations to study the long-term evolution and deployment of charging stations under various policy incentives utilizing complex network theory. These studies provide a theoretical basis for applying complex network theory to study renewable energy trading.

In this paper, renewable energy trading strategies based on evolutionary game theory are proposed. The remainder of this paper is organized as follows: Section 2 gives the mathematical model of the tripartite evolutionary game for renewable energy considering coal-fired power generation enterprises, renewable energy power generation enterprises, and users. Then, in Section 3, a detailed mathematical derivation and conditions are introduced. Next, an actual case of the Guangxi electricity market in China is used to demonstrate the effectiveness and feasibility of the proposed method. Finally, Section 5 presents the paper's conclusion.

## 2. Mathematical Modeling of Tripartite Evolutionary Game for Renewable Energy Participation in the Electricity Market

### 2.1. Evolutionary Game Model

Evolutionarily stable strategy (ESS) was first posited in the 1970s by American biologists Maynard Smith and Price. The basis of ESS is that when a population is affected by mutation strategy, but still able to maintain its original stable state, mutation groups cannot change the original population's state. Therefore, in the process of evolution of a large population, if the above conditions are met, the population can maintain a stable state or stable operation—this state is termed "evolutionary stability". The mathematical description is as follows: suppose there is a population strategy $c$, and the corresponding mutation strategy is $c'$; both strategies belong to the same population $C$, and the function U is used to represent the gain function under different strategies. In summary, when the strategy is an ESS, the conditions satisfied by strategy $c$ are as follows:

$$U(c,c) \geq U(c',c) \tag{1}$$

In Equation (1), it can be seen that if strategy $c$ is adopted and $c \neq c'$, its gain must be greater than its adoption of the variant strategy, the system reaches a steady state at strategy $c$, and strategy $c$ is an ESS.

The replication dynamic equation (continuous form) can be described by the following differential equations:

$$\frac{dx}{dt} = x(t)_i[U(c_i x_i) - U_{aveA}] \tag{2}$$

where $U_{ave} = \sum_{i=1}^{i=n} p_i U(c_i x_i)$. The population is described by an average gain function at moment $t$.

For determining the stability of a solution to a replicated dynamic equation, the equation can be constructed as a Jacobian matrix, and the stability of the solution to the equation can be determined by means of the Lyapunov stability criterion [24,25]. If the real parts of the Eigen roots of the constructed matrix are all negative, then the point is a stable point, i.e., it is an ESS as described previously, and it is consistent with a strict Nash equilibrium. If all the real parts of the eigenvalues of the Jacobian matrix at the point are determined to be positive, then the point is determined to be a point of disequilibrium, which is also referred to as an unstable point, according to Lyapunov; if all the real parts of the eigenvalues of the Jacobian matrix at the point are determined to be both positive and negative, then it is a saddle point.

## 2.2. Participants in the Game

It can be assumed that three main game entities can be set for the process of renewable energy participation in market transactions: coal-fired power generation enterprises (NE), renewable energy power generation enterprises (TE), and users with renewable energy consumption responsibility (EC). In the process of market trading, all types of entities possess limited information and have bounded rationality. They dynamically adjust their trading strategies during the trading decision-making process. Therefore, evolutionary game models can be used for simulation analysis for the participation of these three types of entities in the market trading process.

## 2.3. Game Strategy for Renewable Energy Participation in Guangxi Electricity Market

Regarding renewable energy participation in the market game, due to the inability of all parties to fully grasp market information, the decision-making game process is a dynamic evolution process and involves bounded rationality. Based on the characteristics of various entities in the power market participating in the transaction process, in order to construct an evolutionary game model, parameters are first defined in Table 1.

**Table 1.** Definition of parameters.

| Parameter | Definition |
|---|---|
| $x, y, z$ | Decision space of NE, TE, EC |
| $P_H / P_L$ | The price at which NE participates in transactions at high/low prices |
| $P_N$ | Settlement price for TE not participating in electricity trading |
| $P_G$ | TE green power trading price |
| $C(NE)$ | Cost of NE |
| $C(TE)$ | Cost of TE |
| $Q_N$ | The amount of online electricity that TE does not participate in forelectricity trading |
| $Q_H$ | EC's demand for electricity at high transaction prices |
| $Q_L$ | EC's demand for electricity at low transaction prices |

Note. The above is used as a non-standard parameter table in the model building presented below.

## 2.4. Income Matrix for Renewable Energy Participation in Market-Oriented Transactions

Based on the assumptions and relevant parameter settings in the previous section, as well as the operating rules of the Guangxi electricity market, a benefit matrix for the evolutionary game of NE, TE, and EC can be constructed. The specific situation is shown in Table 2.

**Table 2.** Evolutionary game benefit matrix for TE, NE, and EC.

| | Strategy | EC Purchases Green Electricity $z$ | EC Does Not Purchase Green Electricity $1 - z$ |
|---|---|---|---|
| NE adopts high price $x$ | TE participates in electricity trading $y$ | $(Q_H - Q_G)P_H - C_{NE}$ <br> $Q_G P_H + Q_G P_G - C_{TE}$ <br> $-Q_H P_H - Q_G P_G$ | $(Q_H - Q_G)P_H - C_{NE}$ <br> $Q_G P_H - C_{TE}$ <br> $-Q_H P_H - F$ |
| | TE does not participate in electricity trading $1 - y$ | $Q_H P_H - C_{NE}$ <br> $Q_N P_N + Q_G P_G - C_{TE} - Q_H P_H - Q_G P_G$ | $Q_H P_H - C_{NE}$ <br> $Q_N P_N - C_{TE}$ <br> $-Q_H P_H - F$ |
| NE adopts low price $1 - x$ | TE participates in electricity trading $y$ | $(Q_L - Q_G)P_L - C_{NE}$ <br> $Q_G P_L + Q_G P_G - C_{TE}$ <br> $-Q_L P_L - Q_G P_G$ | $(Q_L - Q_G)P_L - C_{NE}$ <br> $Q_G P_L - C_{TE}$ <br> $-Q_L P_L - F$ |
| | TE does not participate in electricity trading $1 - y$ | $Q_L P_L - C_{NE}$ <br> $Q_N P_N + Q_G P_G - C_{TE}$ <br> $-Q_L P_L - Q_G P_G$ | $Q_L P_L - C_{NE}$ <br> $Q_N P_N - C_{TE}$ <br> $-Q_L P_L - F$ |

Note. $z$ = EC purchases green electricity; $1 - z$ = EC does not purchase green electricity.

From Table 2, it can be seen that NE, TE, and EC jointly form eight evolutionary game strategies. The different decision-making behaviors of different game subjects will have an impact on the game behaviors of the other two types of subjects. The specific situation is as follows:

Strategy 1: NE reports a high price, TE participates in electricity trading, and EC purchases green electricity. The income of NE is the income from participating in electricity energy trading minus the cost of power generation. The income of TE is the sum of the revenue from participating in electricity trading and the revenue from selling green electricity minus its power generation cost. The cost of EC is the sum of participating in electricity trading and purchasing green electricity.

Strategy 2: NE reports a high price, TE participates in electricity trading, and EC does not purchase green electricity. TE has no revenue from selling green electricity. EC does not need to pay for the cost of purchasing green electricity but needs to pay for the assessment of consumption capacity.

Strategy 3: NE reports a high price, TE does not participate in electricity trading, and EC purchases green electricity. Unlike in Strategy 1, NE's electricity-trading capacity is equal to EC's demand capacity. The income of TE is the sum of its non-participating online income and the income from selling green electricity minus the cost of power generation. The cost and strategy of EC are consistent.

Strategy 4: NE reports a high price, TE does not participate in electricity trading, and EC does not purchase green electricity. Therefore, NE returns and strategies are consistent. TE is consistent with Strategy 3, except that it has no green electricity revenue. EC benefits are consistent with those obtained in Strategy 2.

Strategies 5 through 8 are consistent with Strategies 1–4, except for differences in the price and electricity distribution coefficient of electric energy trading.

## 3. Analysis of Evolutionary Game Theory Modeling

### 3.1. Stability Analysis of Evolutionary Strategies for Coal-Fired Power Generation Enterprises

According to the income matrix of coal-fired power generation enterprises (NE) in the previous section, it can be assumed that the expected income of NE participating in transactions at high prices is $U_{NE1}$, the expected income of NE participating in transactions at low prices is $U_{NE2}$, and the average expected income is $U_{NEave}$:

$$U_{NE1} = \begin{aligned} &yz[(Q_H - Q_G)P_H - C_{NE}] + y(1-z)[(Q_H - Q_G)P_H - C_{NE}] + z(1-y)(Q_H P_H - C_{NE}) \\ &+ (1-y)(1-z)(Q_H P_H - C_{NE}) \end{aligned} \tag{3}$$

$$U_{NE2} = \begin{aligned} &yz[(Q_L - Q_G)P_L - C_{NE}] + y(1-z)[(Q_L - Q_G)P_L - C_{NE}] + z(1-y)(Q_L P_L - C_{NE}) \\ &+ (1-y)(1-z)(Q_L P_L - C_{NE}) \end{aligned} \tag{4}$$

$$U_{NEave} = xU_{NE1} + (1-x)U_{NE2} \tag{5}$$

Combining the above equations, the replication dynamic equations of NE are shown in Equations (6)–(8):

$$F(x) = \frac{dx}{dt} = x(U_{NE1} - U_{NEave}) = x(1-x)[(P_L - P_H)Q_G y + Q_H P_H - Q_L P_L] \tag{6}$$

$$\frac{dF(x)}{dx} = (1-2x)[(P_L - P_H)Q_G y + Q_H P_H - Q_L P_L] \tag{7}$$

$$G(y) = (P_L - P_H)Q_G y + Q_H P_H - Q_L P_L \tag{8}$$

With $P_L < P_H$,

$$\frac{dF(x)}{dx} = (1-2x)[(P_L - P_H)Q_G y + Q_H P_H - Q_L P_L] \tag{9}$$

According to the stability theorem of the differential equation, if the probability of NE choosing to participate in the transaction with a high price is in a stable state, it must

satisfy $F(x) = 0$ and $\frac{dF(x)}{dx} < 0$. Since $\frac{\partial G(y)}{dy} < 0$, $G(y)$ is a decreasing function. When $y = y^* = \frac{Q_H P_H - Q_L P_L}{(P_H - P_L)Q_G}$, $G(y) = 0$, at this point $\frac{dF(x)}{dx} = 0$, $F(x) = 0$, and all $x$ values are in an evolutionarily steady state. When $0 < y < y^* < 1$, $G(y) > 0$, then at time $x = 1$ (reported high price to participate in the transaction), $\frac{dF(x)}{dx} < 0$, which is the evolutionarily stable strategy of NE. Conversely, if $0 < y^* < y < 1$, $G(y) < 0$, at this time, at time $x = 0$ (offers a low price to participate in the transaction), $\frac{dF(x)}{dx} < 0$ is the evolutionarily stable strategy of NE. The phase diagram of the evolutionary game theory model is shown in Figure 1.

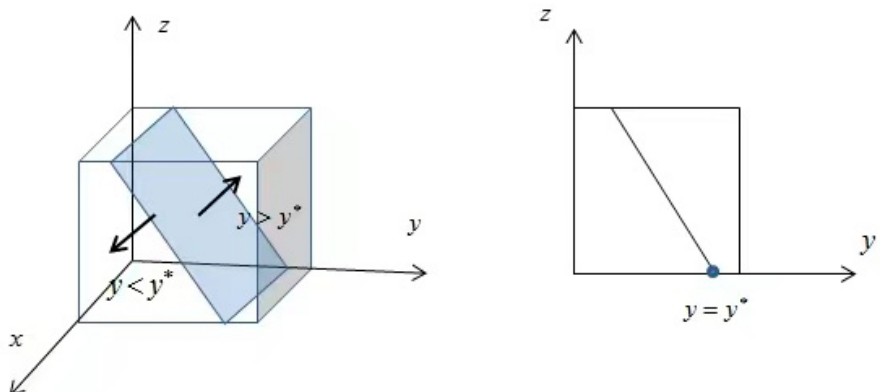

**Figure 1.** Phase diagram of NE evolutionary game theory model.

*3.2. Stability Analysis of Evolutionary Strategies for Renewable Energy Power Generation Enterprises*

According to the income matrix of renewable energy power generation enterprises (TE) in the previous section, it can be assumed that the expected income of TE participating in electricity-market-oriented transactions is $U_{TE1}$, the expected income of TE not participating in electricity-market-oriented transactions is $U_{TE2}$, and the average expected income of TE is $U_{TEave}$:

$$
\begin{aligned}
U_{TH1} = \quad & xz[Q_G P_H + Q_G P_G - C_{TE}] + x(1-z)(Q_G P_H - C_{TE}) \\
& + z(1-x)[Q_G P_L + Q_G P_G - C_{TE}] + (1-x)(1-z)(Q_G P_L - C_{TE})
\end{aligned}
\tag{10}
$$

$$
\begin{aligned}
U_{TH2} = \quad & xz[Q_N P_N + Q_G P_G - C_{TE}] + x(1-z)(Q_N P_N - C_{TE}) \\
& + z(1-x)[Q_N P_N + Q_G P_G - C_{TE}] + (1-x)(1-z)(Q_N P_N - C_{TE})
\end{aligned}
\tag{11}
$$

$$
U_{TEave} = y U_{TE1} - (1-y) U_{TE2}
\tag{12}
$$

Combining the above equations, the replication dynamic equation of TE is shown in Formula (13):

$$
F(y) = \frac{dy}{dt} = y[U(TE)_1 - U(TE)_{ave}] = y(1-y)[Q_G(P_H - P_L)x + Q_G P_L - Q_N P_N]
\tag{13}
$$

Here, $T_1 = 0$, $T_2 = Q_G(P_H - P_L)$, $T_3 = 0$, $T_4 = Q_G P_L - Q_N P_N$.

$$
\frac{dF(y)}{dy} = (1-2y)[Q_G(P_H - P_L)x + Q_G P_L - Q_N P_N]
\tag{14}
$$

$$
G(x) = Q_G(P_H - P_L)x + Q_G P_L - Q_N P_N
\tag{15}
$$

$$
\frac{\partial G(x)}{dx} = \frac{Q_N P_N - Q_G P_L}{Q_G(P_H - P_L)}
\tag{16}
$$

According to the stability theorem of the differential equation, if the probability that TE chooses to participate in the transaction is in a steady state, it must satisfy $F(y) = 0$ and $\frac{dF(y)}{dy} < 0$.

**Assumption 1:** *If $Q_G P_L > Q_N P_N$ (the benefit of renewable energy when it participates in trading at low prices of coal-fired power plants is greater than its benefit when it does not participate in trading), then $\frac{\partial G(x)}{dx} < 0$; therefore, $G(x)$ is a decreasing function. When $x = x^* = \frac{Q_N P_N - Q_G P_L}{Q_G(P_H - P_L)}$, $G(x) = 0$, at this time $\frac{dF(y)}{dy} = 0$, $F(y) = 0$, and all y values are in the state of evolutionarily stable strategy. At time $0 < x < x^* < 1$, $G(x) > 0$, and at time $y = 1$ (participating in the trade), $\frac{dF(y)}{dy} < 0$, which is an evolutionarily stable strategy for TE. Conversely, if $0 < x^* < x < 1$, $G(x) < 0$, at time $y = 0$ (not involved in trading), $\frac{dF(y)}{dy} < 0$, which is the evolutionarily stable strategy of TE.*

**Assumption 2:** *If $Q_G P_L < Q_N P_N$ (the benefit of renewable energy when coal-fired power plants participate in trading at a low price is less than the benefit when they do not participate in trading), then $\frac{\partial G(x)}{dx} < 0$; therefore, $G(x)$ is an increasing function. At time $x = x^* = \frac{Q_N P_N - Q_G P_L}{Q_G(P_H - P_L)}$, $G(x) = 0$, and at time $\frac{dF(y)}{dy} = 0$, $F(y) = 0$, all y are in an evolutionarily stable state. When $0 < x < x^* < 1$, $G(x) < 0$ a, and at time $y = 0$ (not participating in the trade), $\frac{dF(y)}{dy} < 0$, which is the evolutionarily stable strategy of TE. Conversely, if $0 < x^* < x < 1$, $G(x) > 0$, and at time $y = 1$ (involved in trading), $\frac{dF(y)}{dy} < 0$, which is the evolutionarily stable strategy for TE. Figure 2 shows the phase diagram.*

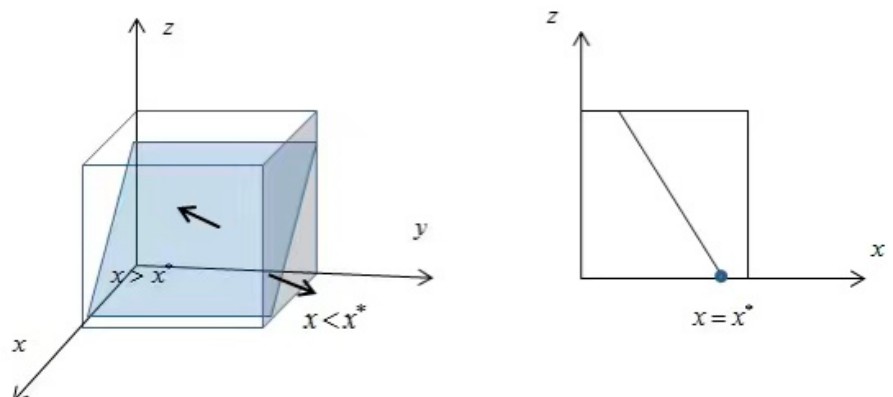

**Figure 2.** Phase diagram of TE evolutionary game theory model.

### 3.3. Stability Analysis of Market User Evolution Strategies

According to the user (EC) benefit matrix of renewable energy consumption responsibility presented in the previous section, it can be assumed that the expected benefit of EC purchasing green electricity is $U_{EC1}$, the expected benefit of EC not purchasing green electricity is $U_{EC2}$, and the average expected benefit for EC is $U_{ECave}$:

$$U_{EC1} = \begin{aligned} & xy[-Q_H P_H - Q_G P_G] + x(1-y)[-Q_H P_H - Q_G P_G] \\ & + y(1-x)[-Q_L P_L - Q_G P_G] + (1-x)(1-y)[-Q_L P_L - Q_G P_G] \end{aligned} \tag{17}$$

$$U_{EC2} = \begin{aligned} & xy(-Q_H P_H - F) + x(1-y)(-Q_H P_H - F) \\ & + y(1-x)(-Q_L P_L - F) + (1-x)(1-y)(-Q_L P_L - F) \end{aligned} \tag{18}$$

$$U_{ECave} = zU_{EC1} - (1-z)U_{EC2} \tag{19}$$

Combining the above equations, the replication dynamic equation for EC is shown in Formula (20):

$$F(z) = \frac{dz}{dt} = z[U(\text{EC})_1 - U(\text{EC})_{ave}] = z(1-z)(-Q_G P_G + F) \tag{20}$$

Here, $E_1 = 0$, $E_2 = 0$, $E_3 = 0$, $E_4 = -Q_G P_G + F$.

$$\frac{dF(z)}{dz} = (1-2z)(-Q_G P_G + F) \tag{21}$$

$$G(x) = -Q_G P_G + F \tag{22}$$

According to the stability theorem of the differential equation, if the probability that TE chooses to participate in the transaction is in a steady state, it must satisfy $F(z) = 0$ and $\frac{dF(z)}{dz} < 0$.

**Assumption 3:** *If $Q_G P_G > F$, then $G(x) < 0$; therefore, only in $z = 0$ (not buying green power), $\frac{dF(z)}{dz} < 0$, which is in an evolutionarily stable state, is the evolutionarily stable strategy for EC.*

**Assumption 4:** *If $Q_G P_G < F$, then $G(x) > 0$; therefore, only in $z = 1$ (buying green power), $\frac{dF(z)}{dz} < 0$, which is in an evolutionarily stable state, is the evolutionarily stable strategy of EC.*

*3.4. Stability Analysis of Evolutionary Strategies for Equilibrium Points in Tripartite Systems*

According to differential Equations (6), (13) and (20), the dynamic equations of (23) are replicated simultaneously:

$$\begin{cases} \frac{dx}{dt} = x(1-x)[(P_L - P_H)Q_G y + Q_H P_H - Q_L P_L] \\ \frac{dy}{dt} = y(1-y)[Q_G(P_H - P_L)x + Q_G P_L - Q_N P_N] \\ \frac{dz}{dt} = z(1-z)(-Q_G P_G + F) \end{cases} \tag{23}$$

The stability of the replicated dynamic equation is determined based on the Lyapunov stability criterion. That is, if the real parts of all eigenvalues of the Jacobian matrix are negative (the determinant value is positive, and the sum of diagonal values is negative), then this point is the stable point of the evolutionary game, which is the ESS mentioned earlier and which conforms to the strict Nash equilibrium state. Based on the replication dynamic equation mentioned above, the Jacobian matrix *J* is constructed:

$$J = \begin{pmatrix} (1-2x)[(P_L-P_H)Q_G y + Q_H P_H - Q_L P] & x(1-x)(P_L-P_H)Q_G & 0 \\ y(1-y)[Q_G(P_H-P_L)] & (1-2y)[Q_G(P_H-P_L)x + Q_G P_L - Q_N P_N] & 0 \\ 0 & 0 & (1-2z)(-Q_G P_G + F) \end{pmatrix} \tag{24}$$

With $\frac{dx}{dt} = 0, \frac{dy}{dt} = 0, \frac{dz}{dt} = 0$, eight pure strategy points—including $x, y, z \in \{(0,0,0)(0,0,1)(0,1,0)(1,0,0)(0,1,1)(1,0,1)(1,1,0)(1,1,1)\}$—can be obtained, as well as two mixed strategy points, (x*, y*, 0) and (x*, y*, 1). By introducing the above values into the Jacobian matrix *J*, the determinant and trace values can be obtained as det (*J*) and tr (*J*), respectively, as shown in Table 3.

$$x^* = -\frac{Q_G P_L - Q_N P_N}{Q_G(P_H - P_L)} \tag{25}$$

$$y^* = -\frac{Q_H P_H - Q_L P_L}{Q_G(P_H - P_L)} \tag{26}$$

$$\alpha = \frac{\sqrt{[(Q_G - Q_H)P_H - (Q_G - Q_L)P_L] \times (Q_G P_L - Q_N P_N) \times (Q_G P_H - Q_N P_N) \times (Q_H P_H - Q_L P_L)}}{Q_G P_H - Q_G P_L} \tag{27}$$

$$\beta = \frac{\sqrt{[(Q_G - Q_H)P_H - (Q_G - Q_L)P_L] \times (Q_G P_L - Q_N P_N) \times (Q_G P_H - Q_N P_N) \times (Q_H P_H - Q_L P_L)}}{Q_G P_L - Q_G P_H} \qquad (28)$$

$$\varepsilon = F - Q_G P_G \qquad (29)$$

**Table 3.** The determinant values, trace values, and eigenvalues of the Jacobian matrix at each stable equilibrium point for NE, TE, and EC.

| Equilibrium | det (*J*) | tr (*J*) | $\lambda_1 \ \lambda_2 \ \lambda_3$ |
|---|---|---|---|
| (0,0,0) | $(Q_H P_H - Q_L P_L) \times (Q_G P_L - Q_N P_N)$ $\times (F - Q_G P_G)$ | $(Q_H P_H - Q_L P_L) + (Q_G P_L - Q_N P_N)$ $+ (F - Q_G P_G)$ | $(Q_H P_H - Q_L P_L), (Q_G P_L - Q_N P_N),$ $(F - Q_G P_G)$ |
| (0,0,1) | $(Q_H P_H - Q_L P_L) \times (Q_G P_L - Q_N P_N)$ $\times (Q_G P_G - F)$ | $(Q_H P_H - Q_L P_L) + (Q_G P_L - Q_N P_N)$ $+ (Q_G P_G - F)$ | $(Q_H P_H - Q_L P_L), (Q_G P_L - Q_N P_N),$ $(Q_G P_G - F)$ |
| (0,1,0) | $[(Q_H - Q_G)P_H - (Q_L - Q_G)P_L]$ $\times (Q_N P_N - Q_G P_L)$ $\times (F - Q_G P_G)$ | $[(Q_H - Q_G)P_H - (Q_L - Q_G)P_L]$ $+ (Q_N P_N - Q_G P_L)$ $+ (F - Q_G P_G)$ | $[(Q_H - Q_G)P_H - (Q_L - Q_G)P_L],$ $(Q_N P_N - Q_G P_L),$ $(F - Q_G P_G)$ |
| (1,0,0) | $(Q_L P_L - Q_H P_H) \times (Q_G P_H - Q_N P_N)$ $\times (F - Q_G P_G)$ | $(Q_L P_L - Q_H P_H) + (Q_G P_H - Q_N P_N)$ $+ (F - Q_G P_G)$ | $(Q_L P_L - Q_H P_H), (Q_G P_H - Q_N P_N),$ $(F - Q_G P_G)$ |
| (0,1,1) | $[(Q_H - Q_G)P_H - (Q_L - Q_G)P_L]$ $\times (Q_N P_N - Q_G P_L)$ $\times (Q_G P_G - F)$ | $[(Q_H - Q_G)P_H - (Q_L - Q_G)P_L]$ $+ (Q_N P_N - Q_G P_L)$ $+ (Q_G P_G - F)$ | $[(Q_H - Q_G)P_H - (Q_L - Q_G)P_L],$ $(Q_N P_N - Q_G P_L),$ $(Q_G P_G - F)$ |
| (1,0,1) | $(Q_L P_L - Q_H P_H) \times (Q_G P_H - Q_N P_N)$ $\times (Q_G P_G - F)$ | $(Q_L P_L - Q_H P_H) + (Q_G P_H - Q_N P_N)$ $+ (Q_G P_G - F)$ | $(Q_L P_L - Q_H P_H), (Q_G P_H - Q_N P_N),$ $(Q_G P_G - F)$ |
| (1,1,0) | $[(Q_L - Q_G)P_L - (Q_H - Q_G)P_H]$ $\times (Q_N P_N - Q_G P_H)$ $\times (F - Q_G P_G)$ | $[(Q_L - Q_G)P_L - (Q_H - Q_G)P_H]$ $+ (Q_N P_N - Q_G P_H)$ $+ (F - Q_G P_G)$ | $[(Q_L - Q_G)P_L - (Q_H - Q_G)P_H],$ $(Q_N P_N - Q_G P_H),$ $(F - Q_G P_G)$ |
| (1,1,1) | $[(Q_L - Q_G)P_L - (Q_H - Q_G)P_H]$ $\times (Q_N P_N - Q_G P_H)$ $\times (Q_G P_G - F)$ | $[(Q_L - Q_G)P_L - (Q_H - Q_G)P_H]$ $+ (Q_N P_N - Q_G P_H)$ $+ (Q_G P_G - F)$ | $[(Q_L - Q_G)P_L - (Q_H - Q_G)P_H],$ $(Q_N P_N - Q_G P_H),$ $(Q_G P_G - F)$ |
| (x*,y*,0) | $\alpha \times \beta \times \varepsilon$ | $\alpha + \beta + \varepsilon$ | $\alpha, \beta, \varepsilon$ |
| (x*,y*,1) | $-\alpha \times \beta \times \varepsilon$ | $\alpha + \beta - \varepsilon$ | $\alpha, \beta, -\varepsilon$ |

According to the determinant values, trace values, and eigenvalues of the Jacobian matrix of NE, TE, and EC at each stable equilibrium point obtained in Table 3, the behavior of each market entity is analyzed as follows: The game evolution behavior of other market entities under different game strategies is analyzed to obtain the decision-making behavior of different decision-making entities.

By analyzing the positivity and negativity of the values of the determinant, trace, and Eigen root of each equilibrium point in Table 3, it is possible to summarize them into sets of data that are positive and negative to each other: (0,0,0) (0,0,1) are positive and negative to each other, (0,1,0) (0,1,1) are positive and negative to each other, (1,0,0) (1,1,1) are positive and negative to each other, (x*,y*,0) (x*,y*,1) are mutually positive and negative. For the above conditions, the following assumptions can be made:

**Condition 1:** $Q_H P_H < Q_L P_L$; *that is, the return of NE when reporting a high price is less than the return of NE when reporting a low price.*

**Condition 2:** $Q_H P_H > Q_L P_L$; *that is, the return of NE when reporting a high price is greater than the return of NE when reporting a low price.*

**Condition 3:** $Q_G P_L < Q_N P_N$; *when TE participates in the trading process and NE reports a low price, the electricity sales revenue obtained by TE is less than the electricity sales revenue obtained by TE when TE does not participate in the trading.*

**Condition 4:** $Q_G P_L > Q_N P_N$; *when TE participates in the trading process and NE reports a low price, TE's electricity sales revenue is greater than TE's electricity sales revenue when TE does not participate in the trading.*

**Condition 5:** $F < Q_G P_G$; *if EC fails to fulfill its consumption responsibility, it is necessary to pay an assessment fee that is less than the electricity purchase cost required to purchase green electricity.*

**Condition 6:** $F > Q_G P_G$; *if EC fails to fulfill its consumption responsibility, it is necessary to pay an assessment fee that is greater than the electricity purchase cost required to purchase green electricity.*

**Condition 7:** $(Q_H - Q_G)P_H < (Q_L - Q_G)P_L$; *when TE participates in the transaction, the profit obtained by NE from bidding high is less than the profit obtained by NE from bidding low.*

**Condition 8:** $(Q_H - Q_G)P_H > (Q_L - Q_G)P_L$; *when TE participates in the transaction, the returns obtained by NE by bidding high are greater than those obtained by NE bidding low.*

**Condition 9:** $Q_G P_H < Q_N P_N$; *when TE participates in the trading process and NE reports a high price, TE's electricity sales revenue is less than TE's electricity sales revenue when TE does not participate in the trading.*

**Condition 10:** $Q_G P_H > Q_N P_N$; *when TE participates in the trading process and NE reports a high price, TE's electricity sales revenue is greater than TE's electricity sales revenue when TE does not participate in the trading.*

Based on the above, the conditions that need to be met for each equilibrium point to achieve evolutionary stability are shown in Table 4.

**Table 4.** Each equilibrium point satisfies the conditions required for evolutionary stability.

| Equilibrium Point | $\lambda_1\ \lambda_2\ \lambda_3$ | Conditions That Need to Be Met to Achieve Stability | Stable Situation |
|---|---|---|---|
| (0,0,0) | $(Q_H P_H - Q_L P_L), (Q_G P_L - Q_N P_N),$ $(F - Q_G P_G)$ | 1, 3, and 5 | Stable point |
| (0,0,1) | $(Q_H P_H - Q_L P_L), (Q_G P_L - Q_N P_N),$ $(Q_G P_G - F)$ | 1, 3, and 6 | Stable point |
| (0,1,0) | $[(Q_H - Q_G)P_H - (Q_L - Q_G)P_L],$ $(Q_N P_N - Q_G P_L),$ $(F - Q_G P_G)$ | 4, 5, and 7 | Stable point |
| (1,0,0) | $(Q_L P_L - Q_H P_H), (Q_G P_H - Q_N P_N),$ $(F - Q_G P_G)$ | 2, 5, and 9 | Stable point |
| (0,1,1) | $[(Q_H - Q_G)P_H - (Q_L - Q_G)P_L],$ $(Q_N P_N - Q_G P_L),$ $(Q_G P_G - F)$ | 4, 6, and 7 | Stable point |
| (1,0,1) | $(Q_L P_L - Q_H P_H), (Q_G P_H - Q_N P_N),$ $(Q_G P_G - F)$ | 2, 6, and 9 | Stable point |
| (1,1,0) | $[(Q_L - Q_G)P_L - (Q_H - Q_G)P_H],$ $(Q_N P_N - Q_G P_H),$ $(F - Q_G P_G)$ | 5, 8, and 10 | Stable point |
| (1,1,1) | $[(Q_L - Q_G)P_L - (Q_H - Q_G)P_H],$ $(Q_N P_N - Q_G P_H),$ $(Q_G P_G - F)$ | 6, 8, and 10 | Stable point |
| $(x^*, y^*, 0)$ | $\alpha, \beta, \varepsilon$ | 5 | Unstable point |
| $(x^*, y^*, 1)$ | $\alpha, \beta, -\varepsilon$ | 6 | Unstable point |

Based on the above 10 assumptions, the conditions required for the related 10 equilibria to reach a stabilization point can be obtained. It can be seen that for $(x^*, y^*, 0)$ $(x^*, y^*, 1)$, regardless of the conditions, the real part of its carry-over to the Eigen root of the Jacobian matrix is not all negative, and therefore the above mixed-strategy point cannot reach an evolutionarily stable point solution. The other eight pure strategy evolutionarily stable points all have game stable points that satisfy specific conditions.

The (0,0,0) strategy needs to satisfy conditions ①, ③, and ⑤ in order to achieve evolutionary stability; i.e., the revenue of NE when it quotes a high price is smaller than the revenue of NE when it quotes a low price, the revenue of TE from electricity sales when it quotes a low price is smaller than the revenue of TE when it does not participate in the transaction, and the appraisal fee that EC needs to pay for the responsibility of unfulfilled consumption is smaller than the cost of purchasing green electricity. Therefore, after satisfying the above conditions, NE tends to enter the market at a low price to gain a larger amount of electricity sales, TE tends to settle at a fixed price without participating in market-based transactions, and EC tends to accept the appraisal fee without purchasing green power.

For the (0,0,1) strategy to achieve evolutionary stability, it needs to satisfy conditions ①, ③, and ⑥; i.e., the gain of NE when NE quotes a high price is smaller than the gain of NE when NE quotes a low price. The income from the sale of electricity obtained by TE when NE quotes a low price during the process of participating in the transaction is smaller than the income from the sale of electricity when TE does not participate in the transaction. The appraisal fee that needs to be paid for the responsibility of the EC for the unfulfilled amount of elimination is larger than the expenditure required for purchasing green electricity. Therefore, after satisfying the above conditions, NE tends to enter the market at a low price to obtain a larger amount of electricity sales, TE tends to settle at a fixed price without participating in market-based transactions, and EC tends to purchase green power.

For the (0,1,0) strategy to achieve evolutionary stability, conditions ④, ⑤, and ⑦ need to be satisfied; i.e., when TE participates in the trading process and NE quotes a low price, the revenue from electricity sales obtained by TE is greater than the revenue from electricity sales when TE does not participate in the trading process. The appraisal fee that needs to be paid for the responsibility of the unfulfilled amount of consumption by EC is smaller than the cost of power purchase that needs to be expended for EC to purchase green power. When TE participates in the transaction, the revenue gained by NE by quoting a high price is smaller than the revenue gained by NE quoting a low price. Therefore, after satisfying the above conditions, NE tends to enter the market with a low price to gain a larger amount of electricity sold, TE tends to participate in market-based trading, and EC tends to accept the appraisal fee without purchasing green electricity.

For the (1,0,0) strategy to achieve evolutionary stability, conditions ②, ⑤, and ⑨ need to be satisfied; i.e., the gain of NE when it quotes a high price is greater than the gain when NE quotes a low price. The appraisal fee to be paid by EC for not fulfilling the responsibility of the consumed quantity is less than the cost of purchasing green electricity. The revenue from the sale of electricity obtained by TE during the process of participating in the transaction when NE quotes a high price is less than that obtained by TE when it does not participate in the transaction. The revenue from the sale of electricity is smaller than the revenue from the sale of electricity when TE does not participate in the transaction. Therefore, after satisfying the above conditions, NE tends to enter the market with a high price, TE tends not to participate in the market-based transaction, and EC tends to accept the appraisal fee without purchasing green power.

For the (0,1,1) strategy to achieve evolutionary stability, conditions ④, ⑥, and ⑦ need to be satisfied; i.e., when TE participates in the trading process and NE quotes a low price, the revenue gained by TE from selling electricity is greater than the revenue gained by TE from selling electricity when it does not participate in the trading process. The assessment fee to be paid by EC for not fulfilling the responsibility for the amount of elimination is greater than the cost of purchasing green electricity. When TE participates in the transaction, the revenue gained by NE quoting a high price is smaller than the revenue gained by NE quoting a low price. Therefore, NE tends to enter the market at a low price, TE tends to participate in market-based transactions, and EC tends to purchase green power.

For the (1,0,1) strategy to achieve evolutionary stability, conditions ②, ⑥, and ⑨ need to be satisfied; i.e., NE's gain when quoting a high price is greater than NE's gain

when quoting a low price. The appraisal fee that EC needs to pay for the responsibility of unfulfilled elimination volume is greater than the cost of EC purchasing green power. The power sales revenue that TE receives when NE quotes a high price is less than that TE receives when it does not participate in the trading process. NE tends to enter the market with a high price, TE tends not to participate in the market-based transaction, and EC tends to purchase green power.

The (1,1,0) strategy needs to satisfy conditions ⑤, ⑧, and ⑩ in order to achieve evolutionary stability; i.e., the appraisal fee that EC needs to pay for the responsibility of unfulfilled consumption is less than the cost of EC purchasing green power. When TE participates in trading, the revenue gained by NE by quoting a high price is greater than the revenue gained by NE quoting a low price. When TE participates in trading, the revenue gained by TE from selling electricity when NE quotes a high price is greater than the revenue gained by TE from selling electricity when TE does not participate in trading. NE tends to enter the market with a high price, TE tends to participate in market-based trading, and EC tends to accept the appraisal fee without purchasing green power.

The (1,1,1) strategy to achieve evolutionary stability needs to satisfy conditions (6), (8), and (10); i.e., EC's need to pay the appraisal fee for not fulfilling the responsibility of the consumption volume is greater than the cost of EC purchasing green electricity. When TE participates in trading, the revenue gained by NE by quoting a high price is greater than the revenue gained by NE quoting a low price. When TE participates in trading, the revenue gained by TE by selling electricity when NE quotes a high price is greater than the revenue gained by TE by selling electricity when TE does not participate in the trading. NE tends to enter the market at a high price, TE tends to participate in market-based trading, and EC tends to purchase green power.

*3.5. Main Findings and Implications of Modeling*

This paper focuses on the basic principles and research methods of evolutionary game theory, constructs a typical model, solves and analyzes the processes of a three-party evolutionary game, describes the establishment method of the payoff matrix, solves the replicated dynamic equations and the stability determination process, and clarifies eight strategy sets in the three-party evolutionary game. The conditions of Jacobian matrix stability discrimination are proposed, i.e., the value of the matrix determinant is greater than zero, the value of the trace is less than zero, or the real parts of the characteristic roots are all negative, and the conditions of meeting the stable point, unstable point, and saddle point are finally determined, which provides theoretical and analytical support for the subsequent research and simulation analysis of the game strategies based on the behavior of the power market players.

However, the simulation analysis of the presented analysis of the game behavior of market players needs to be further strengthened, and some indicators and boundary conditions set in the process of constructing the revenue matrix and replicating the dynamic equations are not comprehensive enough, specifically in the areas of failing to take into account the impact of marginal cost increases brought by variable costs [26]. Meanwhile, the transmission and distribution tariffs and benchmark feed-in tariffs are fixed by default, but with the approval of the new round of transmission and distribution tariffs and the different benchmark feed-in tariffs among different power plants, the model should be further modified and improved in light of these policy changes and the related market realities.

**4. Case Study**

The previous section conducted a theoretical analysis of the game behavior of three types of market entities, namely NE, TE, and EC, and simulated the stable points of each evolutionary game. This section incorporates simulation data from various parties in the Guangxi electricity market to simulate the participation of renewable energy in market-oriented trading in Guangxi, simulating the trading behavior of various market entities (i.e.,

NE, TE, and EC) in the electricity market trading process and simulating the conditions for achieving evolutionary stability.

### 4.1. Boundary Condition Setting

The installed renewable energy power generation of the Guangxi power market accounts for the provision of power to a relatively large number of provinces. Because the authors of this paper are more familiar with the power market in Guangxi and have access to first-hand data on power transactions, this paper selects the power market in Guangxi for modeling and case analysis. In terms of market-oriented transaction electricity prices, due to the significant surge in coal prices, the country and local governments have introduced a series of policies to increase the market-oriented transaction prices of coal-fired power plants. At the national level, a series of policies—such as deepening the market-oriented reform of coal-fired power generation grid electricity prices—have been implemented, while at the local level, various implementation measures have been issued—including Guangxi's implementation of market-oriented reform of coal-fired power generation grid electricity prices. The fluctuation range of coal market trading prices has been further clarified, increasing by 15% to as much as 20%. According to the monthly and semi-annual reports of the Guangxi electricity market in 2022, the average market prices and green electricity average trading prices in the Guangxi electricity market were captured. These data are shown in Table 5.

**Table 5.** Price parameters of Guangxi electricity market.

| Price Parameter Type | Price (CNY/MW·h) |
|---|---|
| Market average price | 485.6 |
| Benchmark grid electricity price of coal-fired power plants | 420.7 |
| Market price ceiling | 504.8 |
| Market price floor | 336.5 |
| Wind power and photovoltaic grid connection prices | 420.7 |
| Average price of green electricity trading | 510 |
| Price parameter type | 485.6 |

Based on the above parameters as boundary conditions for evolutionary game simulation analysis, the stability of various players' games under different market conditions was simulated. MATLAB software 2022 was primarily used for simulation analysis in this article.

### 4.2. Stability Analysis

This section analyzes the important parameters that affect NE, TE, and EC in the market game process, simulates whether the stability of the game under different parameter conditions is consistent with the previous text, and quantifies the impact on all parties in the market in the game process. The specific stability analysis is as follows:

**(1)  Analysis of the impact of assessment fees**.

According to the assumption of boundary conditions, it is known that the minimum trading price of NE is $P_L = 336.5$ (USD/MWh), the maximum trading price of NE is $P_H = 504.8$ (USD/MWh), the feed-in price of TE not participating in the transaction is $P_N = 420.7$ (USD/MWh), and the trading price of TE green power is $P_G = 510$ (USD/MWh). It is also assumed that the demand power for single-user trading in the low-price market is $Q_L = 400$ (MW), the demand power in the high-price market is $Q_H = 350$ (MW), the TE non-participation trading energy feed-in power is $Q_N = 20$ (MW), and the TE participation trading green power feed-in power is $Q_G = 30$ (MW).

According to the analysis of tripartite stability, changes in assessment fees will have a certain impact on whether EC purchases green electricity or pays assessment fees. Therefore, based on the assumption of assessment fees paid by market users for unfinished consumption responsibility, the following three scenarios are assumed and brought into

the replication dynamic equation system: *F* = CNY 18,000, *F* = CNY 16,000, and *F* = CNY 14,000. The results of 50 evolutions over time are shown in Figure 3.

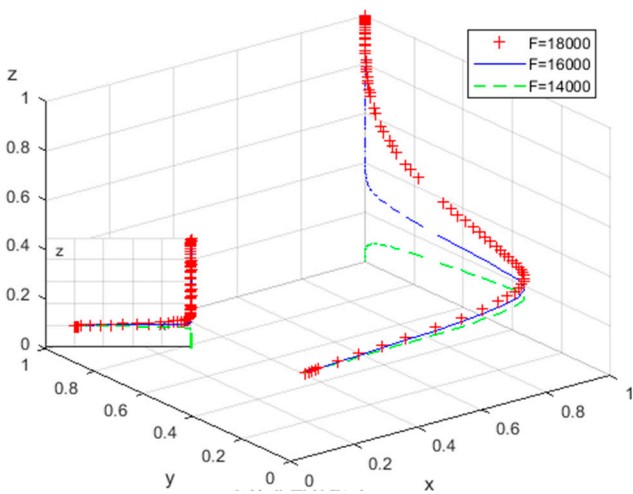

**Figure 3.** The impact of assessment fee *F*.

In Figure 3, it can be seen that under the boundary conditions, as the assessment cost of *F* increases, the probability of EC purchasing green electricity increases, and $Q_G P_G = F = 15,300$ is a critical point. When the assessment cost exceeds 15,300, EC ultimately reaches stability at (1.1.1), and EC tends to purchase green electricity. When F is less than CNY 15,300, EC ultimately reaches stability at (1.1.0), and EC tends to pay assessment fees instead of purchasing green electricity. This is consistent with the analysis of Conditions 5, 8, and 10 and Conditions 6, 8, and 10 in the previous text.

Based on the above parameter settings, we assign values of *F* = CNY 18,000 and *F* = CNY 14,000 and bring them into the replication dynamic equation with 50 evolutions over time. The results are shown in Figures 4 and 5, respectively.

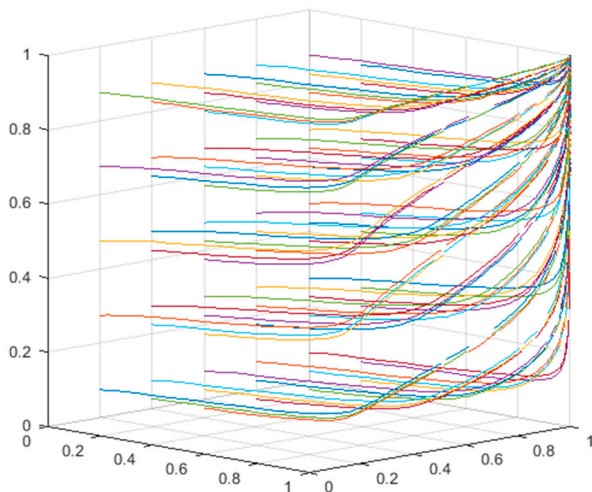

**Figure 4.** Result of 50 evolutions at *F* = CNY 18,000.

Figure 4 verifies that the tripartite evolution will reach a stable point at (1.1.1) when *F* = CNY 18,000.

Figure 5 verifies that the tripartite evolution will reach a stable point at (1.1.1) when *F* = CNY 14,000.

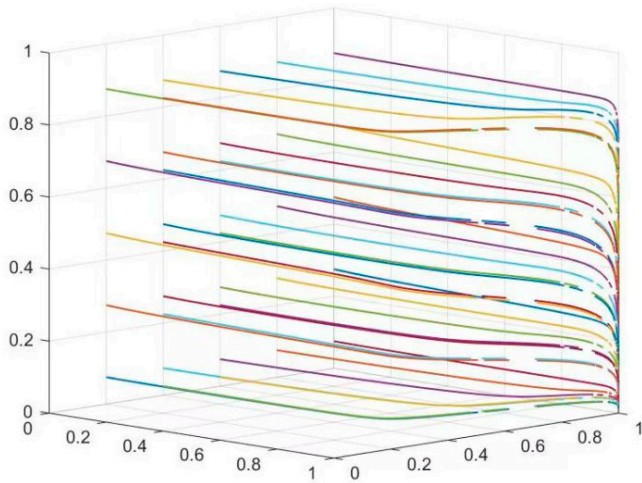

**Figure 5.** Result of 50 evolutions at *F* = CNY 14,000.

**(2)** **The Impact of Different Returns on Coal Fired Power Generation Enterprises Under Different Quotations**

For different price quotes of NE, the main impact on NE revenue is related to EC demand for electricity. According to the demand curve, price and demand are negatively correlated. Therefore, three groups of EC demand for electricity at high and low prices can be set:

Group 1: $Q_L = 400(\text{MW})$ $Q_H = 350(\text{MW})$;
Group 2: $Q_L = 450(\text{MW})$ $Q_H = 350(\text{MW})$;
Group 3: $Q_L = 500(\text{MW})$ $Q_H = 300(\text{MW})$.

The other parameters are still consistent with the previous section. The result of 50 evolutions over time when brought into the replicated dynamic equation system is shown in Figure 6

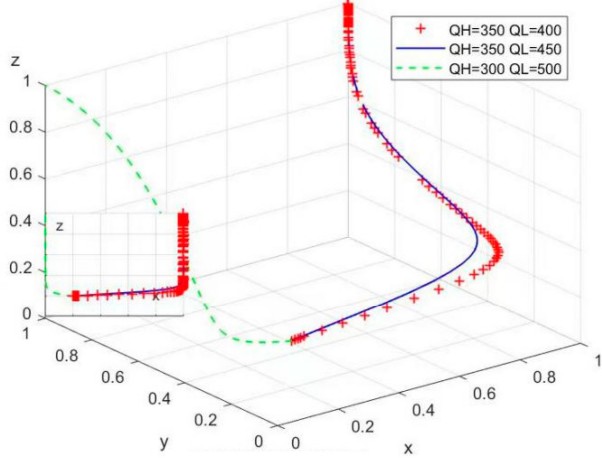

**Figure 6.** The impact of different returns under different quotes by NE.

From Figure 6, it can be seen that under boundary conditions, the stable points of the replication dynamic equation vary with the different price demands of EC at NE. As the difference $P_L Q_L - P_H Q_H$ increases, the probability of NE bidding high increases, with $P_L Q_L = P_H Q_H$ being the critical point. When $P_L Q_L < P_H Q_H$, it reaches stability at (1.1.1), and NE tends to participate in trading at high prices. When $P_L Q_L > P_H Q_H$, it reaches stability at (0.1.1), and NE tends to participate in trading at low prices, which is consistent with the analysis of Conditions 6, 8, and 10 and Conditions 4, 6, and 7 in the above.

Based on the above parameter settings, we assign values to $Q_L = 400(\text{MW})$ $Q_H = 350(\text{MW})$ and $Q_L = 500(\text{MW})$ $Q_H = 300(\text{MW})$ and incorporate these into the replication dynamic equation over 50 times. The results are shown in Figures 7 and 8, respectively.

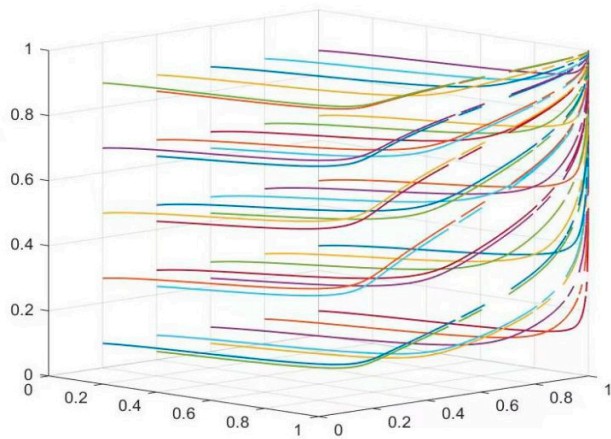

**Figure 7.** Evolution situation with $Q_L = 400(\text{MW})$ $Q_H = 350(\text{MW})$.

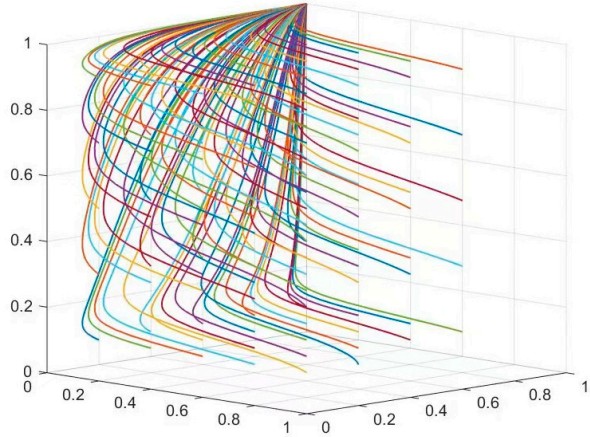

**Figure 8.** Evolution situation with $Q_L = 500(MW)Q_H = 300(MW)$.

Figure 7 verifies that the tripartite evolution will reach a stable point at (1.1.1), when $Q_L = 400(\text{MW})$ $Q_H = 350(\text{MW})$.

Figure 8 verifies that the tripartite evolution will reach a stable point at (0.1.1), when $Q_L = 500(\text{MW})$ $Q_H = 300(\text{MW})$.

**(3) Impact of feed-in tariffs for renewable energy generators not participating in trading.**

According to the assumption of boundary conditions, three sets of TE feed-in price pairs can be set:

Group 1: TE feed-in price without trading (USD/MWh);

Group 2: Feed-in price for TE not involved in trading (USD/MWh);

Group 3: TE feed-in price without trading (USD/MWh).

Since TE's strategy is affected when NE participates in the transaction at high and low prices, NE also sets two control groups, in which NE participates in the transaction at low prices. The results brought into the system of replicated dynamic equations evolved 50 times over time are shown in Figure 9.

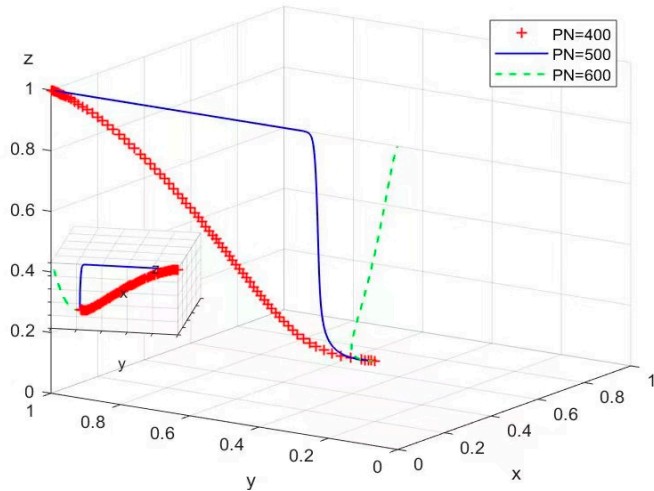

**Figure 9.** Impact of feed-in tariffs when TE does not participate in the transaction.

As can be seen in Figure 9, the replicated dynamic equations stabilize at different points with different $P_N$ prices in the boundary condition case. The critical point is in the low-price market $Q_G P_L = Q_N P_N$; when $Q_G P_L < Q_N P_N$, stabilization is reached at (0.1.1), and TE tends to participate in the trade. At that time, stabilization is reached at (0.0.1), and TE tends not to participate in the trade. This is consistent with the analysis in Section 3.2 above that Conditions 4, 6, and 7 and Conditions 1, 3, and 6 are satisfied.

According to the above parameter settings, we assign values $P_N = 400$ and $P_N = 600$, and the results of bringing these values into the replication of dynamic equations undergoing 50 evolutions over time are shown in Figures 10 and 11, respectively.

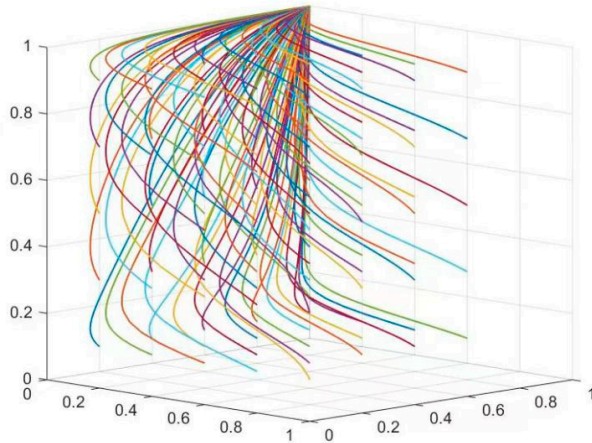

**Figure 10.** Evolutionary situation at $P_N = 400$.

Figure 10 verifies that the temporal tripartite evolution $P_N = 400$ in Figure 9 will reach the stabilization point at (0.1.1).

Figure 11 verifies that the temporal tripartite evolution $P_N = 600$ in Figure 9 will reach the stabilization point at (0.0.1). Therefore, three other groups of TE feed-in price pairs can be set:

Group 4: TE feed-in price without trading $P_N = 400$ (USD/MWh);
Group 5: TE feed-in price without trading $P_N = 600$ (USD/MWh);
Group 6: TE feed-in price without trading $P_N = 800$ (USD/MWh).

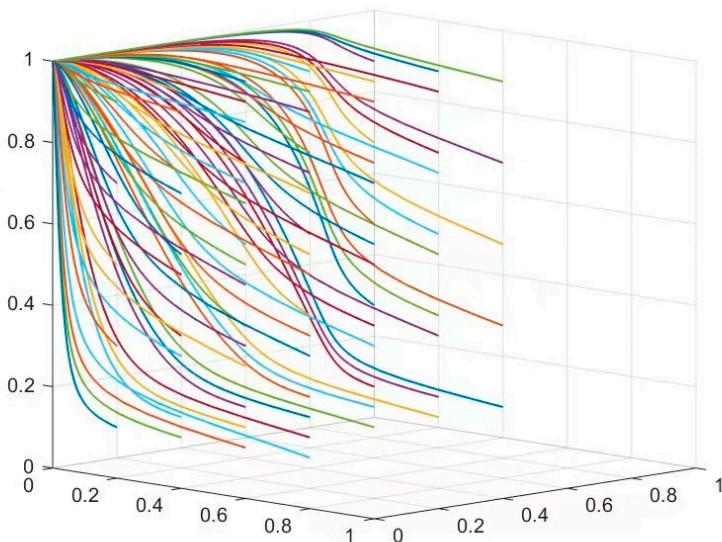

**Figure 11.** Evolutionary situation at $P_N = 600$.

Since TE's strategy is affected when NE participates in the transaction at high and low prices, NE also sets two control groups, which are $Q_L = 400(MW)$ $Q_H = 300(MW)$ (NE participates in the transaction at high prices) and are brought into the replicated dynamic equation set. The results of evolving 50 times over time are shown in Figure 12.

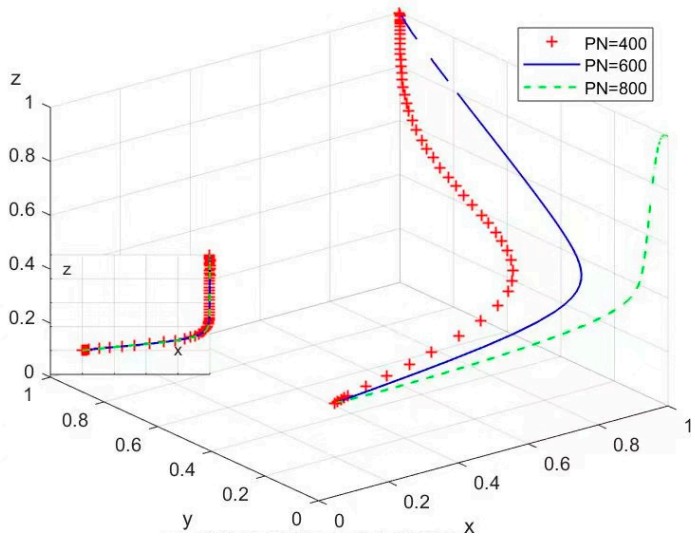

**Figure 12.** Impact of TE's non-participation in trading feed-in tariffs.

As can be seen in Figure 12, the replicated dynamic equations stabilize at different points with different PN prices in the boundary condition case. The critical point is in the high-price market $Q_G P_L = Q_N P_N$; when $Q_G P_L < Q_N P_N$, stabilization is reached at (1.1.1), and TE tends to participate in the trade. When $Q_G P_L > Q_N P_N$, stabilization is reached at (1.0.1), and TE tends not to participate in the trade. This is consistent with the above analysis of satisfying Conditions 6, 8, and 1, and Conditions 2, 6, and 9.

According to the above parameter settings, $P_N = 400$ and $P_N = 800$ are assigned and brought to the replication of dynamic equations over time, and the results of evolving 50 times are shown in Figures 13 and 14, respectively.

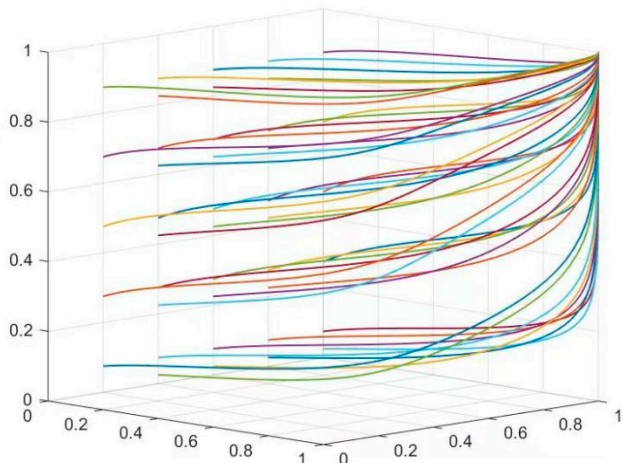

**Figure 13.** Evolutionary situation at $P_N = 400$(three other groups of TE feed-in price).

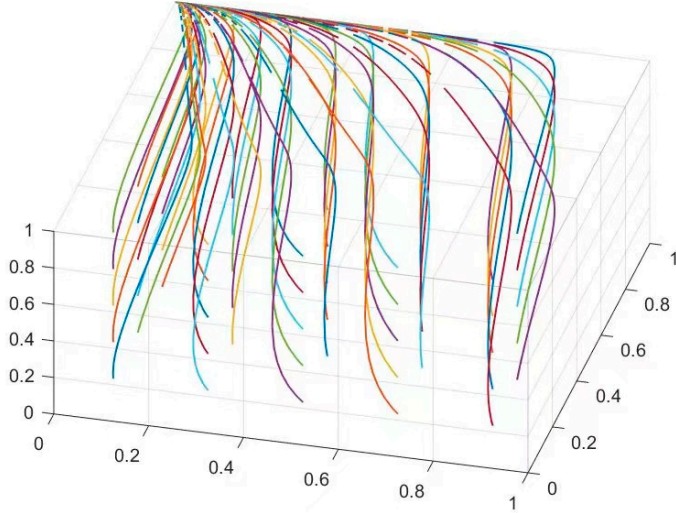

**Figure 14.** Evolutionary situation at $P_N = 800$.

Figure 13 verifies that the temporal tripartite evolution $P_N = 400$ in Figure 12 will reach the stabilization point at (1.1.1).

Figure 14 verifies that the temporal tripartite evolution $P_N = 800$ in Figure 12 will reach the stabilization point at (1.0.1).

**(4)    The impact of green electricity demand on market users.**

Based on the assumption of boundary conditions, three sets of green electricity demand for EC can be set:

Group 1: Green electricity demand of EC $Q_G = 30$ (MW);

Group 2: Green electricity demand of EC $Q_G = 50$ (MW);

Group 2: Green electricity demand of EC $Q_G = 100$ (MW).

To eliminate the impact of NE's participation in high or low prices on stability, it is assumed that the high and low user demand for NE's participation in transactions are both at equilibrium points $Q_L = 450$(MW) $Q_H = 300$(MW), and other parameters remain consistent with the previous section. The results of 50 iterations over time in the replicated dynamic equation system are shown in Figure 15.

From Figure 15, it can be seen that under boundary conditions, as EC's demand for green electricity varies, the stable points of the replicated dynamic equation vary, which has a certain impact on whether EC purchases green electricity. If $Q_G P_G = F$ is a critical point, when $Q_G P_G < F$, it reaches stability at (0.1.1), and EC tends to purchase green electricity.

When $Q_G P_G > F$, it reaches stability at $(0.1.0)$, and EC tends not to purchase green electricity, which is consistent with the analysis of Conditions 4, 6, and 7 and Conditions 4, 5, and 7 above.

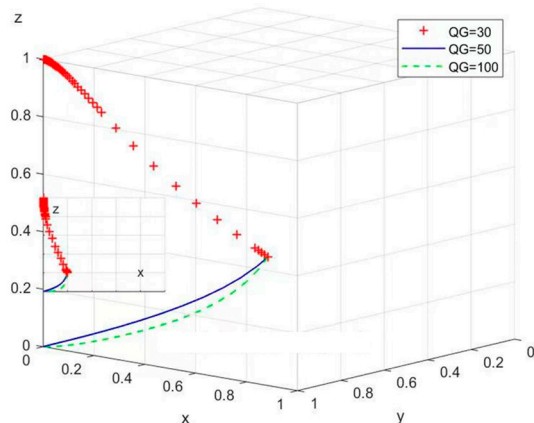

**Figure 15.** The impact of EC green electricity trading demand on electricity consumption.

Based on the above parameter settings, we assign values to $Q_G = 30$ and $Q_G = 100$ and incorporate them into the replication dynamic equation over 50 times. The results are shown in Figures 16 and 17, respectively.

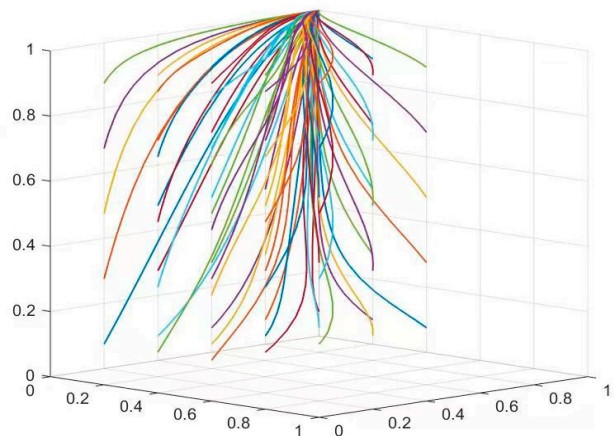

**Figure 16.** Evolution situation with $Q_G = 30$.

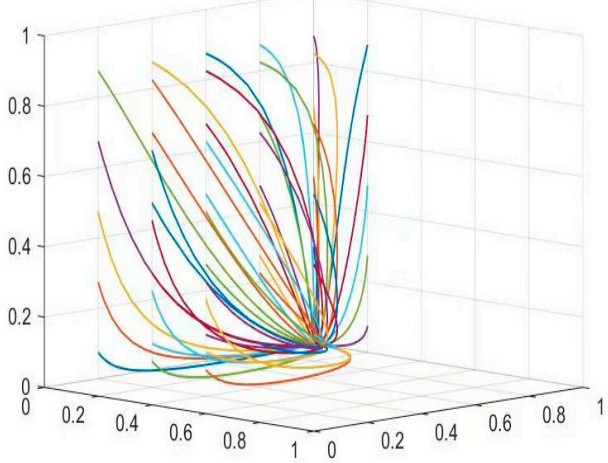

**Figure 17.** Evolution situation with $Q_G = 100$.

Figure 16 verifies that the tripartite evolution will reach a stable point at (0.1.1), when $Q_G = 30$.

Figure 17 verifies that the tripartite evolution will reach a stable point at (0.1.0), when $Q_G = 100$.

### 4.3. Model Simulation Analysis

This section utilized MATLAB tools to simulate the evolutionary game situation under the influence of different parameters, as well as to simulate the change in trading strategies of all parties under different market margin conditions, which provides a model and data support for analyzing the trading behavior of each market player.

The following conclusions are formed: If the appraisal fee is less, it will lead to the users being more inclined to pay the appraisal fee rather than buy green power. Therefore, it is necessary to set the assessment fee for renewable energy consumption reasonably. Next, the difference in market demand will have a greater impact on coal power taking a different offer; the magnitude of the elasticity of electric power demand shows an inverse relationship with the price of coal-fired electricity. The willingness of TE to participate in market-oriented transactions and feed-in price is inversely proportional to the need to set a reasonable benchmark feed-in price for renewable energy to stimulate new energy to participate in the market.

### 4.4. Behavioral Analysis of Decision-Making in Three-Party Bidding Evolutionary Games and Conclusions

In this section, MATLAB tools are applied to simulate the NE, TE, and EC evolutionary game under the influence of different parameters, as well as to simulate the decision-making behaviors of NE, TE, and EC for high or low price participation in the transaction, and to simulate whether or not to participate in the transaction under different parameter settings. In combination with numerical analysis, 10 hypotheses are analyzed, and the following conclusions are drawn:

(1) Analysis of factors influencing whether EC purchases green power. According to the conditions of different assessment fees, it will have an impact on whether EC will eventually buy or not buy green power. If the appraisal fee is small and the penalty for enterprises failing to achieve the consumption weight target is not high enough, EC will choose to pay the assessment fee rather than purchase green power. According to different green power trading price conditions, setting the assessment fees for power grids, power sales enterprises, and power consumption enterprises with consideration of clean energy consumption assessment weights can influence the decision to purchase green power and consume clean energy. According to Figure 3, the larger the assessment fee, the greater the likelihood that green power will be purchased in the trading process. Therefore, local governments and regulators need to set reasonably high assessment fees for the failure to consume a high enough percentage of clean energy, and regulators also need to increase the enforcement of such penalties for enterprises in order to achieve the goals of carbon reduction and renewable energy consumption. This will in turn create a fair and transparent clean energy consumption mechanism in the market and ensure the efficient and orderly consumption of clean energy.

(2) The price of NE participation in the transaction is affected by market supply and demand. According to the different price conditions of the market, the difference in market demand will have a greater impact on NE taking a different offer. According to the "2022 Guangxi Electricity Market Trading Implementation Plan", power generation enterprises adopt the market-oriented feed-in tariff mechanism of "benchmark price + up and down fluctuation", so the maximum price and minimum price of NE in the process of participating in the transaction is limited to a 20% up and down fluctuation of the benchmark price. Therefore, the earnings of NE in different price conditions are more sensitive to the market demand. This means that NE's revenue

under different price conditions is more sensitive to market demand. In the case of tight power supply, the market is not sensitive to the price of electricity, and the difference between the changes in energy demand under different electricity prices is small. In this scenario, the elasticity of demand for electricity is also small, in which case NEs are more inclined to quote higher prices (due to the high cost of coal and the high cost of power generation using coal). In the case of a loose power supply and insufficient start-up rate of large industries, such as under the influence of an epidemic, downward commodity prices, and other factors, the demand side continues to be weak and the demand for electricity is more elastic. Under these conditions, NE is more inclined to quote low prices to stimulate enterprises to increase production and use electricity.

(3) Impact analysis on the situation of whether TEs participate in trading or not. The benefits of TE participation or non-participation in trading are related to feed-in tariffs. According to Figure 12, it can be seen that the willingness of TEs to participate in market-based trading is inversely proportional to the feed-in price of green power. This means that the higher the feed-in price of green power, the higher the probability that TEs will not participate in trading. Therefore, if the feed-in price for renewable energy power enterprises is set too high or subsidized too much, it is not conducive to the entry of renewable energy—such as wind power and PV—into the market. It is necessary to reasonably set the feed-in price of new energy enterprises. Too high is not conducive to market-based consumption; too low will inhibit the construction and investment in new energy generation. Through the market, the price of consumption is formed to guide the new energy production and consumption and achieve a better allocation of new energy generation resources.

## 5. Conclusions

### 5.1. Main Conclusions

This paper constructs an evolutionary game model for coal-fired power plants, renewable energy power plants, and power users using data and assessment weights as set in the Guangxi region. It constructs income matrices for different entities, replicates dynamic equations, and analyzes the decision-making situations of various types of market entities under different conditions. The MATLAB tool was used to simulate the evolutionary game situation under the influence of different parameters, to simulate the trading strategy changes of various parties under different market marginal conditions, and to provide model and data support for analyzing the trading behavior of various market entities. This paper presents the following recommendations:

(1) Building and improving the trading system of the electricity market. In order to effectively respond to the different game strategies of different market entities, it is necessary to establish a more standardized and comprehensive market mechanism. Through the connection of multiple types of trading varieties in different time dimensions, the medium- to long-term market, spot market, capacity market, and auxiliary services market must all be integrated into a single, unified market entity. This will promote the maximum consumption of renewable-energy-market-oriented trading methods. In addition, adjusting the daily and intraday balance deviation through the spot market can address the volatility and intermittency of renewable energy output, such as that of wind power and PV. The benefits of traditional energy can be stabilized through regulatory services, while the costs and benefits of the system can be balanced and guided by renewable energy, based on the principle of equal rights and responsibilities. By establishing a capacity market mechanism, we can ensure the safe operation of the system with a high proportion of renewable energy access, which provides theoretical support for domestic and global power market renewable energy consumption.

(2) Establishing trading varieties that meet the needs of market-oriented support. Due to the different game strategies of traditional coal-fired power plants, renewable energy

producers, and users, different transaction boundary conditions—such as price and electricity quantity—will affect market trends. Therefore, on the basis of the existing green electricity trading mechanism, through the innovation of market-oriented trading varieties, we can enrich the participation of renewable energy in electricity-market-oriented trading, such as clean energy export trading, new energy and thermal power bundling trading, and cross-provincial and cross-regional clean energy trading, effectively balancing the contradiction between traditional energy development and clean energy consumption and expanding the space for clean energy trading. This also provides empirical support for the subsequent optimization of the market mechanism and promotion of the market-based consumption of renewable energy.

(3) Continuously enhancing technical support for market-oriented transactions. In order to better guide the participation of renewable energy sources in the Guangxi electricity market, we will develop more trading varieties that are suitable for clean energy consumption on the basis of existing trading varieties, and increase the development of existing technology support systems to meet the diversified trading experience of market entities. At present, new energy storage can participate in the electricity market as independent energy storage. Based on policy guidance, various entities such as virtual power plants and new energy vehicles may also participate in the electricity market in the future. This has created the need for new requirements for transaction frequency, transaction cycle, transaction flexibility, and transaction deviation settlement. As a core hub of market transactions, trading institutions need to rely on digital and intelligent means to (a) improve the power market information technology support system to adapt to the participation of various entities in trading, (b) provide good trading service support, and (c) carry out timely research on the layout of key trading technologies to promote the consumption of clean energy that relies on technological means.

*5.2. Limitations and Recommendations for Future Research*

In this study, because the market data available for analysis are not perfect, the data mining and model assumptions during the simulation of the decision-making behavior of each market subject still need to be strengthened. There is also a lack of analysis of the participation of the power sales company in the transaction. Specifically, the construction of the model fails to take into account the transaction countermeasures of the power sales company in which there is a simple simulation of the power sales company and the user as a main game body. Furthermore, the future of the three-party evolution of the game simulation process may need a more in-depth study of the content of the four-party evolution of the game.

Market modeling and mechanism design is a long-term and arduous task, but it is necessary to maintain market stability. With the further increase in global renewable energy installed capacity, the new trading mode will be gradually developed, and the original market mechanism and trading varieties will produce subversive changes, which will have a huge impact on and test the market players who have adapted to the medium- and long-term market, and at the same time, combined with the increase in the new energy storage, electric vehicle networking and other new demands, it is necessary to formulate a more scientific, reasonable, and well-standardized market mechanism, and to form a medium- and long-term + spot + power generation. It is necessary to develop a more scientific, reasonable, standardized, and perfect market mechanism, form medium- and long-term + spot + generation + power right multiple trading varieties, and build a flexible and standardized power market that meets the diversified needs of the market, so as to provide a modest contribution to the service of global climate governance and the promotion of clean low-carbon development.

**Author Contributions:** Conceptualization, A.M.A.; Methodology, F.H., H.F., S.Z.A. and H.M.; Software, F.H., H.F., S.Z.A., A.M.A. and H.W.; Validation, F.H., H.F., S.Z.A. and A.M.A.; Formal analysis, F.H., H.F., Y.S., S.Z.A. and A.M.A.; Investigation, F.H., S.Z.A. and A.M.A.; Resources, F.H., Y.W. and

S.Z.A.; Data curation, H.W.; Writing—original draft, Y.S., Y.W., H.M. and H.W.; Writing—review & editing, Y.S., Y.W., H.M. and H.W.; Project administration, S.Z.A.; Funding acquisition, S.Z.A. All authors have read and agreed to the published version of the manuscript.

**Funding:** This study is supported via funding from Prince Sattam bin Abdulaziz University project number (PSAU/2024/R/1445).

**Institutional Review Board Statement:** Not applicable.

**Data Availability Statement:** Data are contained within the article.

**Conflicts of Interest:** Author Fei Huang, Hua Fan, Yunlong Shang and Yuankang Wei was employed by the company Guangxi Power Exchange Center Co., Ltd. The remaining authors declare that the research was conducted in the absence of any commercial or financial relationships that could be construed as a potential conflict of interest.

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
