# Peer review of "Research on Renewable Energy Trading Strategies Based on Evolutionary Game Theory"

_sustainability, doi:10.3390/su16072671_

Round 1

Reviewer 1 Report

Comments and Suggestions for Authors

This manuscript deals with the construction of electricity markets and the improvement of their mechanisms using game theory. This research can also be useful for the global market. Although the importance of non-fossil value has been advocated, in order for consumers to choose non-fossil energy (renewable energy) in the market, it is necessary to devise mechanisms and devices.

This point is examined by looking at the relationship between three parties: fossil-based energy businesses, renewable energy businesses, and consumers.

However, I have doubts as to whether it can be called Evolutionary Game Modeling. I feel that this is too exaggerated as an expression. It is true that mathematical models are used to study decision-making problems involving multiple actors and interdependent situations of behavior, but the point of verification (what is meant by ``Evolutionary'') is not clear.

This point is in demand, and we would like you to choose either 1) change the title (excluding "Evolutionary") or 2) write more clearly and clearly the meaning of "Evolutionary".

Reviewer 2 Report

Comments and Suggestions for Authors

Research on Renewable Energy Trading Strategies Based on Evolutionary Game Theory

Introduction

1.     The introduction establishes the importance of accelerating renewable energy development and substitution actions. However, it could benefit from a more detailed contextualization within the current global energy scenario. Specifically, it would be helpful to include recent trends, challenges, and advancements in renewable energy trading strategies.

2.     While the introduction hints at the focus on renewable energy development, it does not clearly articulate the study's specific objectives. The manuscript would be strengthened by explicitly stating the research questions or hypotheses it aims to address.

3.     The introduction lacks a clear theoretical framework guiding the study. A brief overview of the key theories or models in evolutionary game theory and how they apply to renewable energy trading would provide a stronger foundation for the study.

4.     The introduction could be improved by incorporating a brief review of existing literature on renewable energy trading strategies. It would help position the study within the broader academic discourse and highlight its contribution to the field.

5.     While the introduction establishes the general importance of renewable energy, it lacks a compelling justification for why this specific study is needed. Clarifying what gap in knowledge the study addresses or what unique perspectives it offers would strengthen its rationale.

6.     A brief preview of the methodology, particularly how evolutionary game theory is applied to renewable energy trading strategies, would give readers a clearer understanding of what to expect in the subsequent sections.

7.     The introduction does not specify the expected contributions or implications of the study. Outlining the anticipated findings or potential impact of the research could enhance reader interest and underline the significance of the study.

8.     The introduction could be enriched by connecting the study to broader issues such as sustainable development, climate change mitigation, or economic impacts of renewable energy adoption. It would demonstrate the broader relevance of the research.

Mathematical Modeling of Tripartite Evolutionary Game for Renewable Energy Participation in the Electricity Market

9.     The mathematical modeling section appears to be quite technical and complex. While this is understandable given the subject matter, the authors might consider adding more explanatory text or simplified illustrations to make the content more accessible to readers who may not have a strong background in mathematical modeling.

10.  It is not entirely clear why the authors chose this specific modeling approach. Unlike other potential models, a more detailed explanation of the rationale behind selecting a tripartite evolutionary game model would be beneficial.

11.  The model's parameters and underlying assumptions should be clearly and explicitly stated. It will help readers understand the limitations and applicability of the model.

12.  While the section mentions using a case study from the Guangxi electricity market in China to demonstrate the model's effectiveness, it would be helpful to establish a more explicit link between the theoretical model and its real-world implications.

13.  The manuscript should include a thorough validation of the model, possibly through comparison with existing models or empirical data. It would strengthen the credibility of the model and its findings.

14.  Every model has its limitations. The authors should discuss the limits of their model, including any potential biases or scenarios where the model does not hold.

15.  The manuscript would benefit from a more detailed presentation of the mathematical derivations and conditions. It could be included in the main text or as supplementary material.

16.  The chapter should conclude with a summary of the model's key findings and implications. It will help to contextualize the model within the broader scope of the research and its relevance to the field of renewable energy trading.

Evolutionary Game Modeling Analysis

17.  The manuscript could benefit from a more simplified and precise explanation of the key concepts and strategies. It would aid in making the content more accessible to a broader range of readers.

18.  The chapter should more explicitly connect the evolutionary game modeling analysis to the theoretical framework established earlier in the manuscript. It would help readers understand how the modeling analysis fits within the broader context of the study.

19.  The manuscript presents various strategies and their outcomes but could benefit from a deeper explanation of why certain strategies lead to specific outcomes. It includes discussing the assumptions made and how they influence the model's predictions.

20.  A more precise explanation of why specific modeling choices were made would benefit. It includes the reasons behind choosing specific parameters and how they reflect real-world scenarios in renewable energy trading.

21.  A discussion of the limitations and potential biases of the evolutionary game model would strengthen the chapter. It should include any constraints in the model's applicability to different types of electricity markets or renewable energy scenarios.

22.  Ensure that mathematical equations and representations are presented and explained. Any non-standard mathematical symbols or processes should be defined for clarity.

23.  The chapter should conclude with a discussion of the implications of the modeling analysis. It includes what the findings suggest about the behavior of different entities in the renewable energy market and how this might inform future policies or market strategies.

Case Study

24.  The selection of the Guangxi electricity market for the case study is mentioned, but the manuscript could benefit from a more detailed justification of why this particular market was chosen. Clarifying the relevance of this market to the broader study would strengthen the rationale for the case selection.

25.  The chapter describes the use of simulation data to analyze the participation of renewable energy in market-oriented trading. However, providing more detailed information about the simulation methodology would be beneficial, including the specific parameters used, the software or tools employed, and the reasons behind these choices.

26.  There is a mention of incorporating data from various parties in the Guangxi electricity market, but the manuscript should provide more details about these data sources. Discussing the quality, reliability, and limitations of the data would enhance the credibility of the case study.

27.  While the chapter outlines the process of simulating trading behavior and conditions for achieving evolutionary stability, it could be improved by including a more in-depth analysis of the findings. It includes interpreting the results in the context of the broader research questions and theoretical framework.

28.  It would be interesting to compare the simulated outcomes from the case study and the theoretical predictions made in earlier chapters. It would help validate or challenge the theoretical model.

29.  Every case study has its limitations, and acknowledging these is important. The manuscript should discuss the limits of the case study, including any aspects that may not be generalizable to other electricity markets or renewable energy scenarios.

30.  The chapter should conclude with a discussion of the implications of the case study findings. It includes how the findings contribute to understanding renewable energy trading strategies and what they mean for stakeholders in the electricity market.

31.  Ensure the case study is well integrated with the rest of the manuscript. It should contribute to and support the overall research objectives and not be an isolated section.

Conclusion

32.  The conclusion should provide a clear and concise summary of the study's main findings. It includes a recapitulation of the significant results from the theoretical model, the evolutionary game modeling analysis, and the case study.

33.  While the conclusion mentions the study's implications, it would benefit from a more detailed discussion. It should include theoretical implications for evolutionary game theory and practical implications for renewable energy trading.

34.  The conclusion should address how the study met its initial research objectives or hypotheses. It would help in providing closure to the research narrative.

35.  A critical reflection on the limitations of the study is necessary. It includes limitations in methodology, data, and the generalizability of the findings.

36.  The manuscript could be enhanced by including recommendations for future research. It could involve suggesting new research questions from the study, potential methodological improvements, or areas within renewable energy trading that require further exploration.

37.  Given the focus on mathematical modeling, the conclusion should reflect on the robustness and reliability of the model developed. Discussing any aspects of the model that were particularly successful or areas that could be improved would be beneficial.

38.  The conclusion should be clear, concise, and focused. It should avoid introducing new information or concepts that were not covered in the body of the manuscript.

39.  The manuscript could end with a final thought or perspective that emphasizes the significance of the study in the context of the broader challenges and opportunities in renewable energy trading.

40.  The comments mentioned above are particularly important to enhance the quality of this manuscript.

Comments on the Quality of English Language

Moderate editing of English language required.

Reviewer 3 Report

Comments and Suggestions for Authors

Dear Authors, this paper investigates renewable energy trading strategies Based on evolutionary game theory. I want to express my appreciation for the effort you have put into your research. However, there are some comments and suggestions to improve the quality of the manuscript. The comments are as follows:

·         The introduction effectively sets the stage for the research but could benefit from a more concise overview of existing literature. A clearer articulation of the research gap this paper intends to fill would also enhance this section.

·         The review is thorough but tends to be overly detailed in some parts, potentially overwhelming the reader. A more focused review highlighting key findings relevant to this study’s objectives would be beneficial. Suggested literature:

o   https://doi.org/10.1016/j.segan.2020.100392

o   https://doi.org/10.1016/j.egyr.2021.11.231

·         The empirical analysis is robust, but the paper could improve by discussing the data sources more critically, addressing potential biases or limitations in the data used.

·         The results are presented in a detailed manner, but the paper could benefit from a more explicit connection between these results and their implications for renewable energy trading strategies.

·         While the discussion provides insightful interpretations of the findings, it could be enhanced by drawing more explicit connections to broader energy policy and market implications.

·         The conclusion succinctly summarizes the main findings but could be strengthened by offering more concrete recommendations for practitioners or policymakers based on the study's outcomes.

·         The conclusions could benefit from a discussion on the broader implications of these findings for the shared electric vehicle industry or potential challenges in implementing the proposed strategy.

·         It would be beneficial if the conclusions touch upon areas that need further exploration or specific directions for future research.

Reviewer 4 Report

Comments and Suggestions for Authors

Generally speaking, the devotion to the research can be seen in this paper, which is worthy of appreciation as the method of using the branch of game theory that the author proposed in this paper give a better explanation the long-term evolution of renewable energy development from a multi-agent micro decision-making perspective.

As we can see, this study introduces a tripartite evolutionary game model for renewable energy and proposes multiple income matrices under different strategies, analyzing that the market equilibrium points can be achieved by optimizing trading strategies and their optimization status in promoting renewable energy consumption, the result of which is feasible and has its future research values.

From this study, it can be seen that the author aims to improve the trading system of the electricity market by adjusting the daily and intraday balance deviation through the spot market and others so that to establish a capacity market mechanism to ensure the safe operation of the system with a high proportion of new energy access. And the view shown in this paper seems novel and it’s rewarding to go deeper to research.

Overall, the paper shows positive results that the proposed method highlights and demonstrates the development and utilization of renewable energy as a process of technological innovation promotion, which enhances technical support for market-oriented transactions and the development of existing technology support systems, and improving the power market information technology support system that adapts to the participation of various entities in trading.

However, there’s some grammatical or other details in this paper may deserve further review and carefully check. Here are some examples for reference.

1: some of figures are blur. Please use vector image, such as EPS, pdf, svg format rather than png or jpg.

2: There have some expression problems and grammatical mistakes in some sentences. Here a few examples will be listed from your paper for your reference.

3. There are some minor mistakes about punctuation to be carefully checked.

Line 32-33

Accelerating the development and implementing renewable energy substitution actions is an important measure to promote the energy revolution and build a clean…

It would be clearly to be rewritten as follows. “To accelerate development of renewable energy substitution actions is an important measure to promote the energy revolution and build a clean…”

Line38-39

…Analyzing the impact of market, price, policy and other factors on new energy trading has important practical significance for guiding long-term planning…

A mistake of the phrase “important practical significance” that is repeated semanteme. Significance has already included the meaning of important so it would be proper to delete “important” in this sentence.

Line 368

…It constructs income matrices for different entities, replicates dynamic equations…

Adding a word “also” would be better in the sentence because of the context, which means the sentence would be like “it also constructs income matrices for different entities…”

Line 370-371

…The MATLAB tool was used to simulate the evolutionary game situation under the influence of different parameters, simulate the trading strategy changes of…

To make the sentence more concise. Using “used to simulate the evolutionary…of different parameters and the trading strategy…” better than “, simulate…”

Line 374

…Build and improve the trading system of the electricity market…

“Building and improving” instead of “build and improve”.

Line 388

…Establish trading varieties that meet the needs of market-oriented support…

“Establishing trading” instead of “Establish trading”.

Line 377-378

…the medium to long term market, spot market, capacity market, and auxiliary service market should be…

The sentence would be better to be modified to read “… auxiliary service market, all of them should be integrated into a unified market entity…”

Line 382-384

…By improving the market mechanism for auxiliary services, traditional energy obtains stable benefits by providing regulatory services to the system, while renewable energy guides the system to balance costs and benefits based on the principle of equal rights and responsibilities…

The wrong use of the active and passive relationship shown in this sentence. It is unmatched, like, in the relation of “traditional energy obtains benefits” which seems to a kind of weird expression. It would be more appropriate to be rewritten as follows. “… the benefits of traditional energy could be stable by regulatory services providing to the system, while the balance cost and benefits of the system, based on … responsibilities, would be guided by renewable energy…”

Line 385-387

By establishing a capacity market mechanism, ensure the safe operation of the system with a high proportion of new energy access.

A mistake of the sentence structure that lacking of the subject. It would be proper to be used “By establishing a capacity…the safe operation of the system with a high proportion of new energy access can be ensured.”

Line 408

As a core hub of market transactions, trading institutions need to rely on digital and intelligent means, Improve the power market information technology

A mistake of capitalization of the word “Improve” after the comma.

Comments on the Quality of English Language

see above

Round 2

Reviewer 2 Report

Comments and Suggestions for Authors

Research on Renewable Energy Trading Strategies Based on Evolutionary Game Theory

1.     The manuscript employs evolutionary game theory effectively; a deeper integration with existing theories in renewable energy trading could strengthen its foundation. Exploring additional theoretical models or frameworks complementing evolutionary game theory could provide a more robust analysis.

2.     Broaden the literature review to include more recent studies, especially those offering contrasting views or new insights into renewable energy trading strategies. It will help position the manuscript within the current state of research more effectively. Suggested study https://doi.org/10.1155/2023/4911514  

3.     Provide a clearer rationale for the specific evolutionary game model used, including why it's particularly suited to the research questions. Discuss alternative modeling approaches considered and the reasons for not adopting them.

4.     Enhance the transparency around the data used, especially in the case study. Discuss the source, reliability, and any data limitations in greater detail to strengthen the credibility of the analysis.

5.     If possible, strengthen the validation of the evolutionary game model with empirical data. It could involve comparing the model's predictions with real-world outcomes or using data from similar markets to test its assumptions and results.

6.     Discuss the model's limitations more explicitly, including any assumptions that might affect the generalizability of the findings. Offer suggestions on how future research could address these limitations.

7.     Deepen the discussion of the practical and theoretical implications of the findings. It could involve a more detailed examination of how the results might influence policymaking, market strategies, or the broader field of renewable energy trading.

8.     Provide more specific recommendations for future research, identifying potential areas where the current model could be expanded or applied to different contexts. Suggest methodological improvements or new questions raised by the findings.

9.     Consider making the manuscript more accessible to a broader audience, including those less familiar with evolutionary game theory. It could involve simplifying the technical language, adding more explanatory text, or including visual aids to clarify complex concepts.

10.  Ensure that the conclusions synthesize the main findings, their implications, and how they contribute to the existing body of knowledge. Avoid introducing new information in the conclusion and focus on summarizing the key points made throughout the manuscript.

Comments on the Quality of English Language

Minor editing of English language required.

Author Response

Modification instructions:

  1. In Section 1.2, similar theories to evolutionary game theory were studied, such as the research and analysis methods of experimental economics, and related research on intelligent reinforcement algorithms based on deep learning. The adaptability and effectiveness of evolutionary game theory to this research were emphasized.
  2. Quoting articles in section 3.5: The Impact of Social Preferences on Supply Chain Performance: An Application of the Game Theory Model(https://doi.org/10.1155/2023/4911514 )Expanding the scope of reference in the consumption of renewable energy in this article.
  3. In section 1.2, the adaptability reasons of evolutionary game theory were emphasized, and the advantages compared to other methods were discussed.
  4. In the first paragraph of Section 4.1, it was added that the data source is from the Guangxi electricity market, and the reliability and accuracy of the data source were explained.
  5. In the second paragraph of section 4.3, a comparison of data analysis was added, which was compared with the trading situation in the real market and can provide certain guidance for future and current market transactions.
  6. In Section 5.2, it is emphasized that future research can consider using more complex models to integrate variables of macroeconomic factors, policy changes, and market participant psychological behavior. Meanwhile, the universality and applicability of the model can be verified through comparative studies across markets and regions. In addition, utilizing the latest data science technologies (such as machine learning) to improve the accuracy and applicability of model predictions is also an effective approach.
  7. In section 1.3, it is proposed that research findings have important theoretical significance for understanding and predicting market trading behavior. This article simulates the different trading behaviors of coal-fired power plants, renewable energy power plants, and market users, constructs the income matrices of the above different entities, establishes a replicated dynamic equation, forms a Jacobian matrix, and solves its eigenvalues. Hypothesis analysis is conducted on the formation conditions of each stable point. Research the game strategy among coal-fired power plants, renewable energy, and users in the context of market-oriented consumption of clean energy.
  8. In Section 5.2, Potential areas where this study can be extended or applied include cross-border market transactions, distributed energy resource transactions, and trading platforms that integrate emerging technologies such as blockchain.
  9. The entire manuscript has been effectively simplified, using charts, flowcharts, and examples to explain the key concepts and operational mechanisms of game theory, making the content more intuitive and easy to understand.
  10. The conclusion sections of each chapter and the final chapter have been revised and improved to clearly reflect the research value and contribution of this article. It is pointed out that the research fills the existing research gap in market-oriented trading and provides effective insights for future market trading behavior.

Reviewer 3 Report

Comments and Suggestions for Authors

Dear Authors, thanks for addressing my comments. No further comments from my side.

Author Response

Thank you for your comments